# Large-scale population disappearances and cycling in the white-lipped peccary, a tropical forest mammal

José M. V. Fragoso[1,2,3]*, André P. Antunes[2,4], Kirsten M. Silvius[5], Pedro A. L. Constantino[4], Galo Zapata-Ríos[6], Hani R. El Bizri[7,8], Richard E. Bodmer[9,10], Micaela Camino[11,12], Benoit de Thoisy[13], Robert B. Wallace[14], Thais Q. Morcatty[8,15], Pedro Mayor[16,17], Cecile Richard-Hansen[18], Mathew T. Hallett[19,20,21], Rafael A. Reyna-Hurtado[22], H. Harald Beck[23], Soledad de Bustos[24,25], Alexine Keuroghlian[26], Alessandra Nava[27], Olga L. Montenegro[28], Ennio Painkow Neto[29], Mariana Altrichter[30]

1 Departamento de Zoologia, Universidade de Brasília, Brasília, DF, Brazil, 2 Instituto Nacional de Pesquisas da Amazônia (INPA/MCTIC), Manaus, Brazil, 3 California Academy of Sciences, San Francisco, California, United States of America, 4 RedeFauna–Rede de Pesquisa em Diversidade, Conservação e Uso da Fauna da Amazônia, Tefé, Amazonas, Brazil, 5 Department of Forest Resources and Environmental Conservation, Virginia Tech, Blacksburg, Virginia, United States of America, 6 Wildlife Conservation Society–Ecuador Program, Quito, Ecuador, 7 Department of Natural Sciences, Manchester Metropolitan University, Manchester, United Kingdom, 8 Instituto de Desenvolvimento Sustentável Mamirauá, Tefé, Amazonas, Brazil, 9 Museum of Amazonian Cultures-Fundamazonia, Iquitos, Loreto, Perú, 10 DICE, School of Anthropology & Conservation, University of Kent, Canterbury, United Kingdom, 11 Proyecto Quimilero, Roosevelt 4344, CABA, Resistencia, Argentina, 12 EDGE of Existence—Zoological Society of London, Regent's Park, London, England, United Kingdom, 13 Kwata NGO, Cayenne, French Guiana, 14 Wildlife Conservation Society, Bronx, New York, United States of America, 15 Department of Social Sciences, Oxford Brookes University, Oxford, United Kingdom, 16 Departament de Sanitat i d'Anatomia Animals, Facultat de Veterinària, Universitat Autònoma de Barcelona, Bellaterra, Spain, 17 Museo de Culturas Indígenas Amazónicas, Loreto, Iquitos, Peru, 18 Office Français de la Biodiversité-DRAS/SCGEE UMR EcoFoG, Kourou, France, 19 Department of Wildlife Ecology & Conservation, University of Florida, Gainesville, Florida, United States of America, 20 Institute for the Environment & Sustainability, Miami University, Oxford, Ohio, United States of America, 21 Center for International Forestry Research (CIFOR), Bogor, Indonesia, 22 El Colegio de la Frontera Sur -Unidad Campeche, Campeche, Campeche, México, 23 Department of Biological Sciences, Towson University, Towson, Baltimore, Maryland, United States of America, 24 Secretaría de Ambiente y Desarrollo Sustentable de Salta, Santiago del Estero, Salta, Argentina, 25 Fundación Biodiversidad Argentina, Suipacha, Argentina, 26 Peccary Project/IUCN/SSC Peccary Specialist Group, Campo Grande, Brazil, 27 Fiocruz ILMD Amazon, Adrianópolis, Manaus, Amazonas, Brazil, 28 Instituto de Ciencias Naturales, Universidad Nacional de Colombia, Bogotá, Colombia, 29 Tropical Sustainability Institute–TSI, Carapicuíba, São Paulo, Brazil, 30 Faculty Environmental Studies, Prescott College, Prescott, Arizona, United States of America

* fragoso1@mac.com, jose.fragoso@unb.br

## Abstract

Many vertebrate species undergo population fluctuations that may be random or regularly cyclic in nature. Vertebrate population cycles in northern latitudes are driven by both endogenous and exogenous factors. Suggested causes of mysterious disappearances documented for populations of the Neotropical, herd-forming, white-lipped peccary (*Tayassu pecari*, henceforth "WLP") include large-scale movements, overhunting, extreme floods, or disease outbreaks. By analyzing 43 disappearance events across the Neotropics and 88 years of commercial and subsistence harvest data for the Amazon, we show that WLP disappearances are widespread and occur regularly and at large spatiotemporal scales



**Data Availability Statement:** All data are included in the article.

**Funding:** None of the authors are reporting funding support for this study.

**Competing interests:** The authors have declared that no competing interests exist.

throughout the species' range. We present evidence that the disappearances represent 7–12-year troughs in 20–30-year WLP population cycles occurring synchronously at regional and perhaps continent-wide spatial scales as large as 10,000–5 million km$^2$. This may represent the first documented case of natural population cyclicity in a Neotropical mammal. Because WLP populations often increase dramatically prior to a disappearance, we posit that their population cycles result from over-compensatory, density-dependent mortality. Our data also suggest that the increase phase of a WLP cycle is partly dependent on recolonization from proximal, unfragmented and undisturbed forests. This highlights the importance of very large, continuous natural areas that enable source-sink population dynamics and ensure re-colonization and local population persistence in time and space.

## Introduction

Periodicity in animal population dynamics—i.e., population cycling—has long been observed in nature in northern vertebrate (small and medium sized mammals, birds, fish) and invertebrate populations [1,2]. In general, cycling populations of vertebrates and insects show rapid decline phases and slow growth phases [2,3]. Non-cyclic population dynamics also occur as random trends in population size, punctual, one-time decreases and increases, or irregular recurring increases and decreases with variable amplitude and periodicity.

The white-lipped peccary (*Tayassu pecari*, henceforth "WLP") is classified as Vulnerable in the IUCN red list and in Appendix II of CITES. This is the only tropical forest ungulate that forms large, permanent, cohesive herds comprising hundreds of individuals [4]. In the Amazon, the home range of a herd of about 400 animals may extend to 200 km$^2$ [4]. Considering an adult weight of 30 to 50 kg, such a herd represents 12,000 to 20,000 kg of biomass moving across the landscape, rooting up soils, consuming seeds, seedlings, plant parts and animal matter, dropping excreta, creating wallows, and providing abundant food for large cats, all processes that influence biotic communities [5] and the carbon cycle [6]. These ecological processes may be of even greater significance given that entire populations of WLP have been documented to disappear and eventually reappear over large geographic areas, with important consequences for forest composition and the livelihoods of forest-dependent peoples [4]. WLP population dynamics are therefore of great interest in studies of forest ecology, as cyclicity in their abundance may permit periodic recruitment of plants and influence the predation pressure exerted by predators such as jaguars (*Panthera onca*) and pumas (*Puma concolor*) on other prey species, including large rodents, other ungulates, xenarthans, and reptiles [7]. Fluctuations in WLP abundance also influence human ecology because, when abundant, WLPs are a very important source of protein for indigenous and rural people, representing up to 70% of hunted biomass in some cases [4,8]. While the WLP's unusual social organization and use of space is increasingly well-studied [9,10], its population fluctuations are poorly understood [4,11].

Cycling can occur in mammal species with high reproductive rates when they are subject to non-linear or over-compensating density-dependent mortality. Even when mortality is linear and density-dependent, cycling can still occur if prolonged negative environmental or other conditions depress reproduction and / or extend high mortality rates into the decline phase of the cycle despite reduced densities [2,3]. Although the conditions necessary for cycling to occur are understood and can be demonstrated mathematically, the specific drivers of cyclicity are diverse and vary within and among species. Endogenous factors include variations in

reproductive output and other inherited qualities [12] or genetic drift [13], while exogenous factors include interactions with predators [14], diseases [15], food availability [16,17], and physical impacts, such as fires and floods [18]. Dispersal dynamics and social behaviors also influence the shape of cycles [2].

Potentially cyclic population dynamics have been documented in small populations of temperate ungulates in fragmented ranges [19], but given the altered nature of these populations, it is difficult to assess whether these represent cases of cycling in the same way documented for snowshoe hares (*Lepus americanus*) [14,17] and small rodents such as lemmings and voles [20], given the reduced size of the populations and the fragmented nature of their range.

In the 1980s, researchers in the Amazon began to report occasional local disappearances of WLP populations, attributing them to migration and other large-scale movements [21,22], overhunting [23], or disease outbreaks [24]. Since then, there have been many documented disappearance and reappearance of populations over large areas [4,11], but these have been documented as isolated events and not evaluated as a species phenomenon. Because WLPs are difficult to study in their remote, dense forest habitats, and due to their large home-range, we fill this gap in knowledge by compiling multiple sources of evidence on fluctuations in WLP abundance throughout the species' range and by analyzing the extent, timing and periodicity of documented disappearances. We also examine over-compensating density-dependent mortality as a possible explanation for the disappearances and recoveries in WLP populations.

## Methods

### Survey

We conducted a questionnaire-based survey of experts from 2016 to 2019 (S1 Table). This questionnaire was sent to 58 Neotropical wildlife researchers, including all members of the IUCN/SSC Peccary Specialist Group. To complement the questionnaire, we conducted a Google Scholar search of the key words white-lipped peccary, *Tayassu pecari* and peccary and extracted the same information as from survey respondents for instances of multiyear, extensive area disappearances. When there was more than one report for the same date and location, these were consolidated as one event, while separate reports from the same or adjacent locations within a year of each other were retained in the data set but counted as single events. Disappearances at sites separated by 100 km were considered distinct events. We estimated the mean years of absence, SD and IQR for cases at sites >100,000-ha that were surrounded by contiguous forest. In cases where a range of time was provided, we used the average of the range.

### Commercial pelt trade and subsistence hunting record

We examined legal pelt trade records from 1932 to 1969 from 379 cargo manifests, port registers and financial documents [25], governmental statistics from Peru between 1946 and 2019 [26], and subsistence hunting records from 66 published and unpublished studies from 1965 to 2017 (S2 Table) from the South-western Amazon. The combined dataset includes records of nearly 2 million WLPs hunted for their hides and meat over an estimated 686-million ha. We controlled for variation in hunting effort by calculating the proportion of WLP pelts and kills to another similarly preferred and priced species, the collared peccary (*Pecari tajacu*). For each year and locality, we divided the number of WLP hunted by the total hunted of the two species (*WLP/WLP+CP)*. The resulting dataset contains 749 WLP hunting-proportion events for both commercial and subsistence purposes. We assume that the number of kills reflects the abundance of the species on the hunting grounds: when the number of hunted WLPs is low relative to CP, the WLP populations are low, individuals are difficult to find and kill, and few pelts

show up in the records. Conversely, if the proportion of WLP to CP is high, we assume that the WLP population is also high. This assumption is supported by encounter rates and kill rates for the two species over 10 years in French Guiana [27]. Using the proportion of hunted WLP to CP allowed the joint analysis of commercial and subsistence hunting data, by successfully controlling differences in hunting effort throughout time and space.

## Population fluctuations analysis

We modeled the fluctuations of the WLP proportion through time as population fluctuations. The unknown proportion of the WLPs hunted in relation to CP in year $t$, for $t$ = 1932 to 2019, denoted by $\gamma_i(t)$, is assumed to be a smooth curve over time. Since the proportion varied between 0 and 1 we used a beta inflated distribution model, $X(t){\sim}BEINF(BI(t),\alpha,\gamma,\mu,\phi)$–a mixture distribution between beta and Bernoulli distributions that allows values 0 and 1 for the response variable–where $BEINF(BI(t))$ denotes a beta inflated distribution with mean $BEINF(BI(t))$, $\mu$ location and $\phi$ scale parameters that control the probability density function, and two extra parameters $\alpha$ and $\gamma$, which model the probabilities at zero and one [28,29]. To track the population fluctuation curve $\gamma_i(t)$ as a cubic spline [30,31], we used gamlss package (Version 5.1–6) [32] in R language software [33], which estimates $\alpha,\gamma,\mu,\phi$ and spline parameters (number and position of knots for the curve $\gamma_i(t)$) by maximum likelihood with automatized Generalised Akaike Information Criteria for model selection [29]. We used the different watersheds as a random factor due to different sample sizes among basins, and to possible non-independence of sampled localities within each basin.

## Results and discussion

### Literature review and expert reporting of disappearances

We recorded 67 reports from experts on 43 independent WLP disappearances at 38 sites in 9 countries (S2 Table), occurring over a cumulative area of at least 50-million ha. Twenty-eight disappearance sites are in contiguous old-growth forest in the Amazon region, nine in forest fragments in Atlantic Forest in southern Brazil and northern Argentina, and one in rainforest in Guatemala (S2 Table).

At small scales (<100,000 ha), we defined a disappearance as a sudden (no more than 12 months elapsed between the last documented normal to superabundant population level (see below) and a documented low or absent level) decline from previously documented normal to superabundance to low abundance or absence. In the absence of intensive searching, it is difficult to distinguish between low abundance and absence over larger areas; with the exception of the French Guiana case described below, such intensive searching has rarely been undertaken. We therefore defined a disappearance as occurring when no or very few WLP were sighted by frequent expert visitors to the area and / or local families and hunters for at least 3 years, and a reappearance as the first known documentation of population levels similar to those that occurred at that site prior to the period of superabundance and crash. Based on recorded abundances across the Amazon, for the purposes of this paper we describe five categories of abundance–zero / absent; low; normal/non-extreme; high; and superabundant–but recognize that low and high densities will differ among sites and that a detailed, continued study of population growth and decline is needed at different site before we can clearly define which densities constitute extremes. Continuous monitoring of WLP densities or abundances using standardized methods over more than two years are very rare. Some studies do include accurate documentation of herd encounter rates and herd size; these abundance measures correlate with density and are more frequently reported. In the Amazon, a density of about 5 individuals per $km^2$ is low about 10 individuals per $km^2$ reflects a non-extreme norm, [9,34]; 15

individuals per km$^2$ is high; while at superabundance, densities can reach 20 to 100 or more individuals per km$^2$ [35]. Not all population declines had been documented continuously; in these cases, the absence periods were estimated by researchers. Large scale (>10,000 km$^2$) disappearances may occur gradually over 2 to 5 years over a contiguous area, with the start of the event progressing outward from one location [27].

Two cases, one in Brazil and one in Guyana, illustrate the densities and encounter rates involved in rapid population crashes (S2 Table). At Maracá Island Ecological Reserve in the northern Brazilian Amazon state of Roraima, WLPs were at a superabundant density (138.8 ind/km$^2$) in 1988 and herds were sighted almost every day during intensive searching [35]. WLPs then ceased to be detected in March 1989 and were not detected again despite continuous monitoring until a small herd (A) of 39 individuals was seen in January 1991(1.4 ind/km$^2$) and a second with 139 individuals (herd B) was observed in September 1991 (8.3 ind/km$^2$) [4]. Herd size grew very slowly and by 2005 herd A contained approximately 100 individuals and B about 400, normal for the Amazon, but still lower than the superabundant numbers just prior to the crash. In Guyana, intensive research over 3.5 years described a WLP population with high to superabundant densities of up to 25 ind/km$^2$ up until June 2010, when the study ended [36,37]. WLP absence was documented starting in 2011 and for the next 7 years by camera trapping, transect surveys and hunter reports for the region (S2 Table). As of 2019, WLP had not reappeared in most areas of Guyana.

Population absences or low abundances for 23 cases in contiguous forest lasted on average 10.3 years (SD = 4.9), with 50% lasting between 7 and 12 years. Some population lows were well documented through long-term field research. For example, in contiguous forest regions of the Amazon biome in French Guiana, WLPs were continuously monitored between 2000 and 2021. They were at easily detectable abundances (encounter rates) on line transects across the territory in 2000 and 2001 [27]. Monitoring revealed a geographically expanding absence starting in 2003, reaching almost complete absence from 2006 to 2009 across the entire region. During 2012 and 2013, WLPs were again detected in several locations [38], and abundance indices (line transect and relative kill rates) continue to increase over the country until present days.

Longer absences are reported from discontinuous forests outside the Amazon. In the 41,704-ha Intervales State Park, located in the largest remnant of Atlantic Forest fragment in São Paulo State, Brazil, WLPs ceased to be detected in 1990 and were again detected in 2017 after a 27-year absence. In the larger Iguaçu National Park, also in Brazil's Atlantic Forest, and adjacent northern Misiones province in Argentina, WLPs ceased to be detected around 1995. Years of surveys confirmed their absence until several individuals were detected with camera traps in 2016, likely immigrating from nearby forested areas in Argentina [39]. In Turvo State Park (Rio Grande do Sul State, Brazil), WLPs ceased to be detected in 1980 and have not been seen since then (S2 Table).

At several sites, more than one disappearance event has been documented through time, pointing to population cyclicity. In and around Manu National Park, Peru, WLPs disappeared in 1978, reappeared 12 years later, then disappeared again in 2011. Twenty individuals were detected in 2015, but as of 2019, the population had not returned to pre-disappearance levels. Two distinct disappearance events occurred in the 275,531-ha Rio Jutaí Extractive Reserve, located in continuous, undisturbed forest in western Amazonas state, Brazil. Local inhabitants reported the start of the first disappearance between 1998–2002, with the timing depending on the location within the reserve. WLPs reappeared between 2006–2011, with the timing again depending on the site (S2 Table). Researchers were present for the second disappearance, which began in 2013 near the mouth of the river (S2 Table). By 2014 WLPs had disappeared from the adjacent section of the watershed and in 2015 from the more distant headwater

region. WLPs have not yet been detected at any location in the reserve. In the eastern part of the state, three independent disappearance events sequential in time were reported in an area of more than 3-million ha, including Amanã Sustainable Development Reserve, Unini River Extractive Reserve and Jaú National Park (S2 Table). In Madidi National Park, Bolivia, WLPs disappeared in 1984 and were absent for 10 to 14 years, then reached pre-disappearance densities before disappearing again in mid-2017 along the Heath River, followed by the Tuichi and Hondo rivers in 2018 (S2 Table).

Because researchers observe a limited set of sites in a region, the true geographical extent of disappearances is usually not known. Nevertheless, we documented several cases of disappearances occurring at large spatial scales: 35% of the disappearances affected areas larger than 1-million ha (S2 Table). In French Guiana, WLPs disappeared over the territory's entire 8-million ha extent (Fig 1). In Guyana, they disappeared over at least 5.4-million ha in the Rupununi region. In the northern Amazon of Brazil, WLPs disappeared over at least 4.8-million ha in Roraima state (Fig 1). In Madidi National Park in Bolivia and Manu National Park in Peru, disappearances affected about 1-million and 1.7-million ha, respectively (S2 Table). In the next section we show that historical pelt records support the large geographical extent of WLP disappearances.

## WLP population fluctuations in the western Amazon documented through hunting returns

We examined legal pelt trade records from 1932 to 2019 [24,25] and subsistence hunting returns from 66 studies between 1965 and 2017 from the western Amazon. The combined dataset includes records of almost 2 million WLPs hunted for their hides and meat over an estimated 686-million ha over 88 years (Fig 1). To detect fluctuations in WLP peccary returns while accounting for spatiotemporal variation in hunting effort, we relied on the relative frequency of hunted WLPs and collared peccaries (CP). The two peccary species are among the species most preferred by subsistence hunters [36,40] and prices for their skins were nearly identical during the period analyzed [25], such that hunting effort is equal for the two species at any one time. By tracking the proportion of hunted WLP relative to total peccaries hunted of both species (WLP/WLP+CP) we control for the differences in overall hunting effort among years while assuming that hunting effort is equal for both species in any one year. Population fluctuations have not been observed or documented for CPs, which live in small, territorial family groups (6 to 20 individuals) [41] rather than large, home-range-overlapping herds like WLPs. We modeled fluctuations in the WLP to CP proportion (WLP/WLP+CP) as a proxy for population fluctuation.

Based on this time series (1932 to 2019), three synchronic, geographically constrained growth and decline events in WLP abundance are evident within the western Amazon (Figs 1 and 2). Two large peaks (1941 and 2006) were followed by rapid declines (late 1940s, and late 2000s), while a third peak (1975) showed a more gradual decline (perhaps due to limited data during this period). Increases were gradual in all cases—population growth required over 20 years to reach population maxima—while declines occurred as fast, large scale population crashes over 5 years, as evidenced by the steeper slopes of at least two of the declines.

The time series of WLP hunting harvests in the western Amazon analyzed here provides strong empirical evidence of population cycling in the species over large geographical areas. WLP population cycles appear to last 20 to 30 years from peak to peak, with a rapid population decline phase over 1 to 5 years, an absence or low-abundance phase of 7 to 12 years, followed by a slow growth phase of about 20 years.

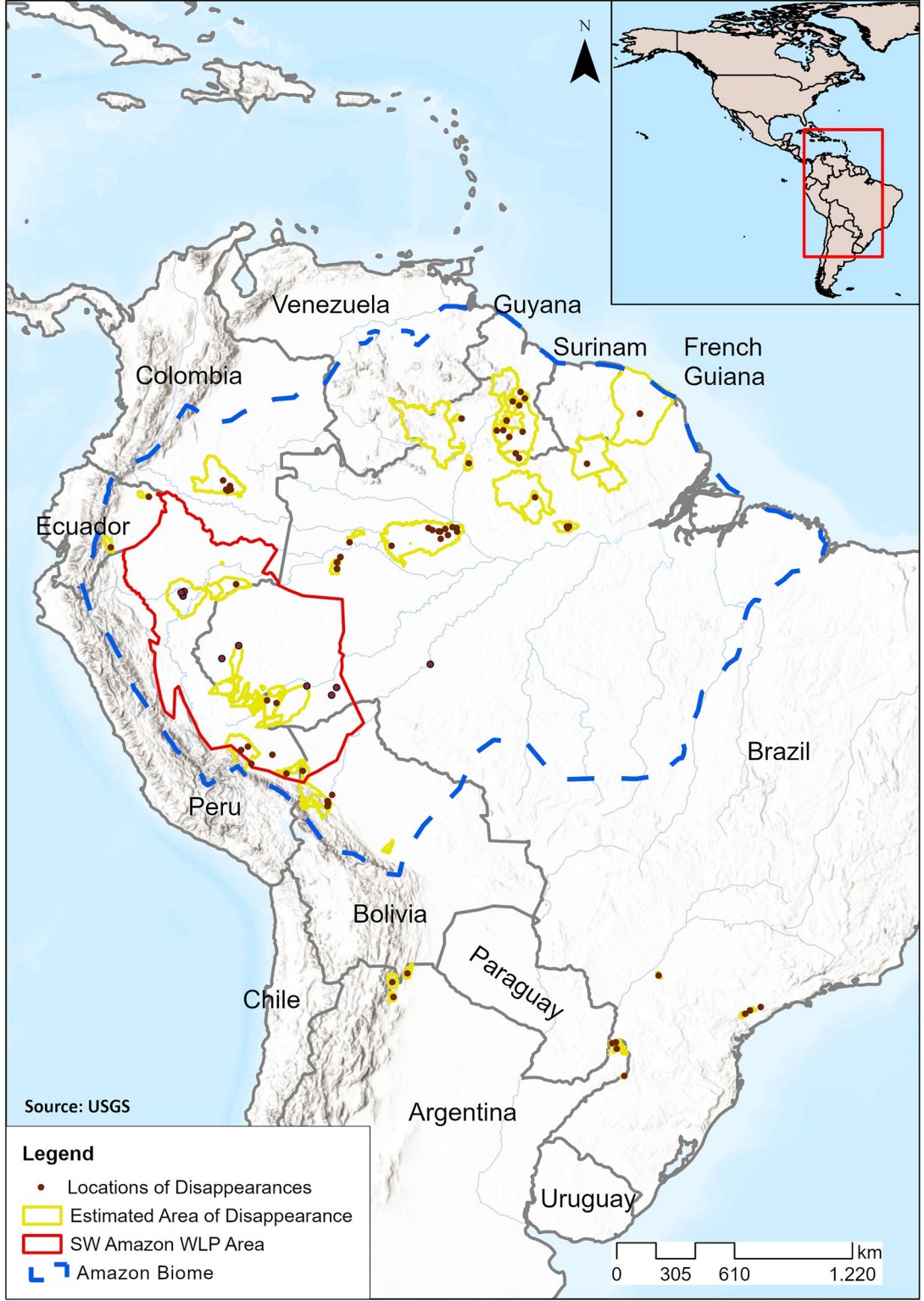

**Fig 1. Locations of white-lipped peccary (*Tayassu pecari*) disappearances.** Brown dots mark disappearances of small or unknown areal extent. Yellow lines mark the estimated extent for disappearances over larger regions. The 686-million ha western Amazon region, the source area for pelt and hunting data, is delineated in red. The blue line demarcates the Amazon biome. The disappearance site in Guatemala is not shown. Base map provided by the United States Geological Survey (USGS).

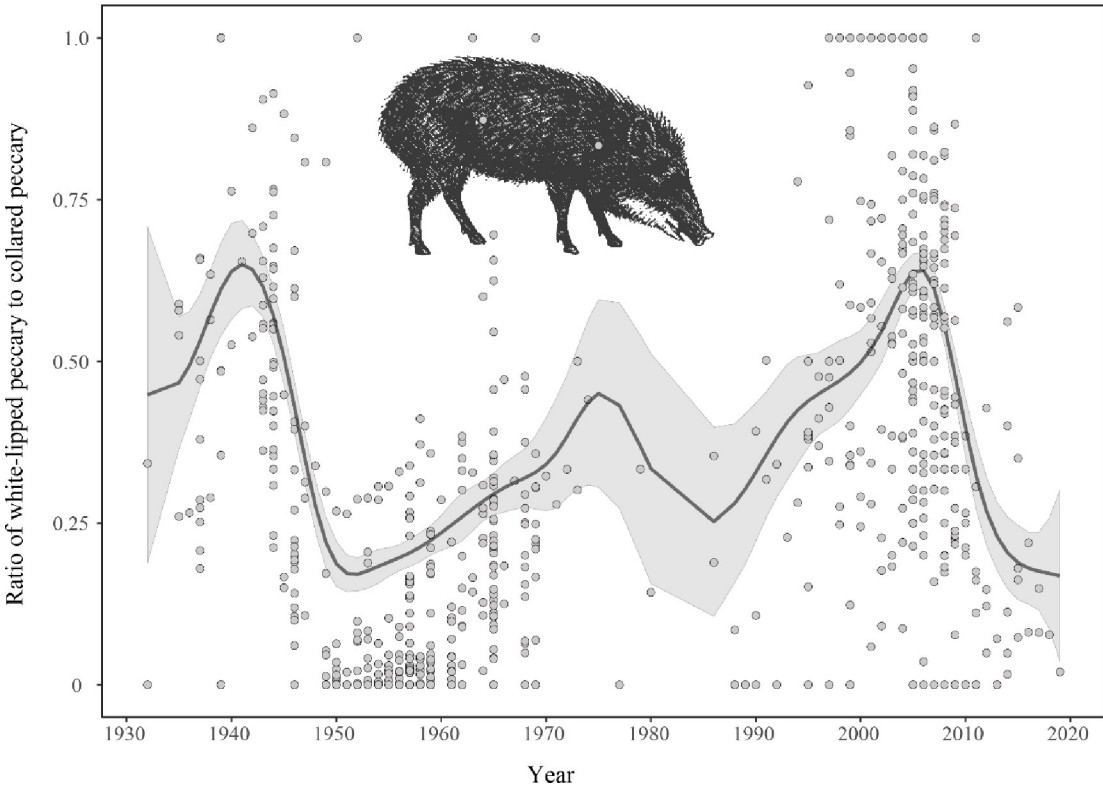

**Fig 2. Estimated population fluctuation of WLP from 1932 to 2019 in the western Amazon.** Data from commercial pelt and subsistence hunting for meat. WLP fluctuations are based on the proportion of hunted WLP to the total of WLP plus hunted collared peccary. The shaded gray area indicates 95% confidence interval. Circles are individual data points.

## Causes of WLP disappearances: Is it a cyclic species?

The enigmatic population dynamics pattern described here is unexpected for any large forest mammal and previously undocumented for any Neotropical mammal. The data on timing and spatial extent of WLP disappearance compiled from multiple sources suggests that the dynamics are periodic (and therefore cyclical) and occur at regional scales of about 1–5 million km$^2$. Such periodicity could be driven by external, periodic drivers, or by intrinsic, density-dependent factors. Several external factors have been suggested in the literature. We review those here but note that none of them can account for periodic, cyclic fluctuations. We propose intrinsic, density-dependent dynamics as an alternative explanation, while noting that some of the external factors could contribute to extended populations lows and, in the case of diseases, may be part and parcel of the density-dependent dynamics.

## Previously proposed drivers: Hunting

Although WLP meat is preferred by local peoples and the species is vulnerable to overhunting [42], many studies have found WLP hunting to be sustainable in the Amazon region [36,37,43]. It is unlikely that the WLP disappearances we documented are caused by hunting: most disappearances occurred in unpopulated, remote areas of contiguous forest; and there is no reason for or indication of historic changes in hunting effort sufficient to drive the boom and crash periods. For instance, in Acre State, Brazil and Loreto Department, Peru, the number of hunters remained relatively constant in the monitored sites during the 2000s and 2010s. Continued reduced survival due to hunting can however amplify population declines caused

by other factors [4]. We expect hunting to be more deleterious in small, isolated forest remnants than in continuous Amazon forest with undisturbed source areas that buffer impacts of local hunting on WLP populations [4,11].

## Previously proposed drivers: Local and regional movements driven by environmental stressors

WLP disappearances have been attributed to long-distance and / or migratory events related to environmental stressors [4,22]. In the western Amazon, WLPs have been observed to move out of flooded forest to non-flooded tierra-firme forest during high floods, sometimes crossing large rivers [18]. In cases of extreme flooding, mortality may be occurring together with these movements. For example, WLPs monitored intermittently since 1990 through a combination of line-transects, hunter interviews and camera trap methods in the Yavari-Samiria region, an area covering 4-million ha of tierra-firme and flooded forests in the Peruvian Amazon [18,44], move seasonally from flooded to upland forests. Local population fluctuations in each of these habitats may be related to intensity of flooding. Within the tierra-firme forests of the Yavari valley, WLPs have shown extreme population fluctuations over 30 years, including a decline from a high of 15 ind./km$^2$ in 2000 to 2.0 ind./km$^2$ in 2004 and a complete disappearance in 2013. In contrast, in flooded forests the WLPs showed less-extreme oscillations of 7.7 ind./km$^2$ +- 3.4 (SD), but also disappeared between 2013 and 2017 following historically high and prolonged floods that began in 2011 and lasted through 2015. This regional disappearance following extreme flooding was reflected in meat sales in the Iquitos bush meat market, which provides a measure of WLP harvest from the Loreto Region, an area of 38-million ha that includes the Yavari landscape [45]. In 1996 there were 4,510 individuals sold, in 2006 3,170 were sold, and then a dramatic decrease was seen in 2013 when only 253 WLP were sold. In 2018 sales rebounded with 3,227 individuals sold. This suggests that population dynamics in the Yavari are shaped by superposition of local-scale between-habitat movements and regional-scale shifts in mortality and reproduction.

Our knowledge of WLP movements is still too incipient to infer WLP disappearances from food-related migratory movements, especially across the very large spatial scales observed. The bearded pig (*Sus barbatus*) of Southeast Asia periodically forms large herds that then move long distances, presumably tracking fruiting by preferred trees [46]. However, the large home ranges used by WLPs afford them with significant within- and between-season food diversity, including fish and invertebrates in drying wetlands and palm fruits and seeds in palm-dominant forest types [4,41].

## Previously proposed drivers: Disease

The gregarious nature of WLPs, their unusually large herd size for a forest dwelling ungulate, and the overlapping home ranges that allow multiple herds to use that same feeding sites and wallows may set the conditions for rapid disease transmission that locally weakens or eliminates entire herds and populations [47]. Emmons [23] proposed that a local disease outbreak caused a WLP population disappearance event in Manu National Park, Peru. In 1997, Fragoso [4,11] postulated epidemic outbreaks at regional scales (millions of ha), similar to those documented for temperate and tropical Old-World ungulates [48,49], after noting that the northern Roraima, Brazil disappearance event was concurrent with a foot and mouth disease outbreak in livestock in adjacent ranchlands. He proposed that disease outbreaks could be an intrinsic and recurrent consequence of the species' unique social organization, herd size and movements. WLP carcasses untouched by predators or humans were found in northern Roraima during this period. Similarly, WLP carcasses were found during unexplained disappearances

in Manu and Tambopata in Peru, and in Guatemala. Diseased but live individuals were also sighted at Viruá National Park in Brazil and in Mexico during population lows.

However, no carcasses were reported in most other events (n = 30) and no causes for the disappearances were proposed in those events. Novel pathogen(s) infecting WLP have not been identified and it is still an open question whether disease outbreaks could drive population collapses, on their own or in synergy with other stressors.

## Newly proposed driver: Density-dependent overcompensation

An alternative explanation for the WLP disappearances is over-compensation in density-dependent responses, with the most likely proximate causes being decreased reproductive rates and higher mortality caused by weakened condition in super-abundant populations. The gregarious nature of WLPs, whose herds reach several hundreds of individuals at population maxima, as well as the relatively high species-specific high reproductive rate (1.64–1.77 young per litter, 158 days of pregnancy, and 18 months age of first reproduction) [50–52] create plausible conditions for exceeding carrying capacity, subsequently triggering over-compensating mortality [19] and negative maternal effects [3], which could explain the rapid crash phase of WLP populations. Limited food, diseases, parasites or any hidden physiological stressor would be amplified by high densities, resulting in rapid crash periods for a species whose population cyclicity is mostly driven by density-dependent abundance and growth. To our knowledge, population cycles have not been documented, studied or confirmed for any Neotropical mammal, and non-cyclic population fluctuations, while occasionally observed [46,53], have not been documented or studied.

The observation that declines are often preceded by super-abundance concurs with a density-dependent explanation. In this scenario, when WLP populations reach high numbers, food is less available per individual, female reproductive productivity decreases, and there is greater infant and adult mortality. The population goes into a fast retraction leading to declines and disappearances. External drivers, including a decline in food or habitat quality independent of population increase, are not necessary explanations in this scenario, but could play a role by depressing population recovery, as could disease affecting weakened animals or extreme drought, or habitat reduction with crowding in the remaining habitat due to flooding. As noted earlier, hunting during a decline phase could further pressure the population but have little or no impact during the increase and stability phase.

Anthropological studies support the hypothesis of cyclicity in WLP populations [56]. The WLP is highly prized as food by indigenous peoples [36] and plays a prominent role in their religious beliefs [54–56]. Although our own data are restricted to recent history, WLP disappearances appear to have occurred for long enough to be incorporated into the cosmologies and beliefs of various indigenous peoples in the Amazon [4,54–56], which suggests that severe population fluctuations are a characteristic of the species. Several ethnic groups describe disappearance events as linked to the death of a powerful shaman; their return can only be secured by another shaman using special cultural activities [4,56].

## Cumulative effects of different factors

It is possible that in some sites, under some circumstances, there are cumulative impacts of different factors such as hunting, environmental stressors, and disease in the population cycle. For example, in the flooded forests in the Peruvian amazon, we see the cumulative impacts of hunting and environmental stresses. When intensive floods occur, WLPs retreat to levees (floodplain islands) where food becomes limited and natural predation increases leading to population reductions. People hunt on the levees where peccaries are trapped which results in

high hunting pressure [18]. This cumulative impact of floods and hunting leads to greater population declines than individually. Fragoso [4] posited that human hunting pressure during periods when populations are low may cause local extinction. In Meso-America and Southern SA research has shown the cumulative impacts of habitat destruction, drought, and hunting [10,42]. Habitat destruction and drought causes reduced food availability and large mammals become concentrated in smaller forest blocks, or around small ponds with greater resource competition, and where hunters have greater success [42]. In the tierra firme forests of western Amazonia studies have shown hunting is generally sustainable and does not cause population declines of WLP with no known cumulative impacts (18, 43–45). In regions that have cumulative impacts (flooded forests, Meso-America, Southern SA) the WLP populations have more severe disappearances and slower recoveries.

## Re-appearances

WLP re-appearance consisting of gradually increasing numbers of individuals may result from population growth from a few remaining individuals, re-colonization after dispersal from adjacent populations at high or superabundant levels or both [4]. Their herding social structure may mean that populations cannot reach high growth rates until herds reach a threshold size; this, together with the need for immigration from other populations to re-establish local populations may be factors that delay recovery. The period of absence or low abundance for WLP in continuous habitat is around 7–12 years at the large local scale and 20 years in the large regional scale. On the other hand, the absence period was over 20 years in sites with high forest fragmentation due to anthropogenic impacts, suggesting that in the absence of dispersal, regular cycling dynamics are interrupted, and a population may go extinct.

Immigration is considered a factor in the growth phase of other species' cycling populations [2]. If immigration is important in WLPs, then this suggests that there should be a lag time in the recovery of adjacent populations. For example, when the WLPs crashed in the Brazilian Yanomami Indigenous territory (Amazon River drainage area [4]; the Yanomami reported that WLPs started returning from over the Parima mountains from Venezuela (Orinoco River drainage area). This implies that the population in Venezuela was in a growth phase, which was driving dispersal of the animals. While we documented population disappearances occurring throughout the Amazon basin, our data indicate that synchronicity in cycling occurs at large, possibly basin-wide scales. This would create a source-sink dynamic across areas with dispersal connecting populations across the entire basin.

## Conservation measures

The large oscillations in WLP populations, caused by different drivers acting independently or cumulatively, make the species vulnerable to local extinctions as global change brings more and more environmental stressors. High forest fragmentation decreases the possibilities of dispersal and repopulation, brings formerly isolated populations of wild species in contact with domestic animals, facilitating disease transmission, and increases hunting pressure on populations now isolated in forest fragments. Therefore, we emphasize the importance of large continuous areas and biological corridors for the maintenance of a WLP metapopulation and source-sink dynamics [57]. In fragmented areas where hunting pressure is high or accumulates with other factors such as drought or flooding, hunting should be regulated and probably stopped during the early years of WLP population recovery. Investments in long-term monitoring of WLPs in continuous, intact forests, through feasible, low-cost approaches such as participatory monitoring [36,43,53], are needed before we can fully describe populations fluctuations and their drivers. More research on forest dynamics while WLPs are in high

abundance and where they are absent is also needed to fully understand the role of this important but enigmatic species in Neotropical forests.

## Supporting information

**S1 Table. Questions emailed to researchers that work with white-lipped peccaries (*Tayassu pecari*).**
(DOCX)

**S2 Table. Reports from experts on 43 independent white-lipped peccaries (Tayassu pecari) disappearances at 38 sites in 9 countries.**
(DOCX)

## Acknowledgments

J.M.V.F. thanks the Yanomami people for documenting the return of WLP to their area in the 1990s, and the Makushi and Wapichan people for tracking their hunting returns so precisely. Thanks also to Roan McNab, Luis Pacheco, Arnaud Desbiez, Anthony Giordano, Mineração Rio do Norte and the Biota Projetos e Consultoria Ambiental for providing information. C.R. H. thanks the Parc Amazonien de Guyane and European Community. Thanks to Bill Magnusson and Albertina Lima for providing a home at INPA for the writing of a first draft of the manuscript.

## Author Contributions

**Conceptualization:** José M. V. Fragoso, André P. Antunes, Cecile Richard-Hansen, Rafael A. Reyna-Hurtado.

**Data curation:** José M. V. Fragoso, André P. Antunes, Kirsten M. Silvius, Pedro A. L. Constantino, Galo Zapata-Ríos, Hani R. El Bizri, Richard E. Bodmer, Micaela Camino, Benoit de Thoisy, Robert B. Wallace, Thais Q. Morcatty, Pedro Mayor, Cecile Richard-Hansen, Mathew T. Hallett, Rafael A. Reyna-Hurtado, H. Harald Beck, Soledad de Bustos, Alexine Keuroghlian, Alessandra Nava, Olga L. Montenegro, Ennio Painkow Neto.

**Formal analysis:** José M. V. Fragoso, André P. Antunes, Kirsten M. Silvius, Pedro A. L. Constantino, Galo Zapata-Ríos, Hani R. El Bizri, Rafael A. Reyna-Hurtado, Mariana Altrichter.

**Investigation:** José M. V. Fragoso, André P. Antunes, Kirsten M. Silvius, Pedro A. L. Constantino, Galo Zapata-Ríos, Hani R. El Bizri, Richard E. Bodmer, Micaela Camino, Benoit de Thoisy, Robert B. Wallace, Thais Q. Morcatty, Pedro Mayor, Cecile Richard-Hansen, Mathew T. Hallett, Rafael A. Reyna-Hurtado, H. Harald Beck, Soledad de Bustos, Alexine Keuroghlian, Olga L. Montenegro, Ennio Painkow Neto, Mariana Altrichter.

**Methodology:** José M. V. Fragoso, André P. Antunes, Benoit de Thoisy, Mariana Altrichter.

**Project administration:** José M. V. Fragoso, Mariana Altrichter.

**Supervision:** José M. V. Fragoso, Mariana Altrichter.

**Validation:** José M. V. Fragoso, André P. Antunes, Kirsten M. Silvius, Micaela Camino.

**Writing – original draft:** José M. V. Fragoso, André P. Antunes, Kirsten M. Silvius, Micaela Camino, Benoit de Thoisy, Mariana Altrichter.

**Writing – review & editing:** José M. V. Fragoso, André P. Antunes, Kirsten M. Silvius, Pedro A. L. Constantino, Galo Zapata-Ríos, Hani R. El Bizri, Richard E. Bodmer, Micaela Camino,

Benoit de Thoisy, Robert B. Wallace, Thais Q. Morcatty, Pedro Mayor, Cecile Richard-Hansen, Mathew T. Hallett, Rafael A. Reyna-Hurtado, H. Harald Beck, Soledad de Bustos, Alexine Keuroghlian, Alessandra Nava, Olga L. Montenegro, Ennio Painkow Neto, Mariana Altrichter.

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
