## [Decision Letter · Decision Letter 0]

26 Apr 2022

PONE-D-21-25351

Disappearances and population cycles of a large mammal in Neotropical forests

PLOS ONE

Dear Fragoso

Thank you for submitting your manuscript to PLOS ONE. After careful consideration, we feel that it has merit but does not fully meet PLOS ONE’s publication criteria as it currently stands. Therefore, we invite you to submit a revised version of the manuscript that addresses the points raised during the review process.

We look forward to receiving your revised manuscript.

Kind regards,

Bilal Habib

Academic Editor

PLOS ONE

Journal Requirements:

5. We note that Figure 1 in your submission contain map images which may be copyrighted. All PLOS content is published under the Creative Commons Attribution License (CC BY 4.0), which means that the manuscript, images, and Supporting Information files will be freely available online, and any third party is permitted to access, download, copy, distribute, and use these materials in any way, even commercially, with proper attribution. For these reasons, we cannot publish previously copyrighted maps or satellite images created using proprietary data, such as Google software (Google Maps, Street View, and Earth). For more information, see our copyright guidelines: http://journals.plos.org/plosone/s/licenses-and-copyright.

Additional Editor Comments (if provided):

First of all I apologies for the extended delay with respect to review process of this paper. The delay was purely because of delayed response from the reviewers. We have finally received one review and based on the review it is recommended to revise your submission.

Reviewers' comments:

Reviewer's Responses to Questions

**Comments to the Author**

1. Is the manuscript technically sound, and do the data support the conclusions?

Reviewer #1: Partly

2. Has the statistical analysis been performed appropriately and rigorously? 

Reviewer #1: Yes

3. Have the authors made all data underlying the findings in their manuscript fully available?

Reviewer #1: Yes

4. Is the manuscript presented in an intelligible fashion and written in standard English?

Reviewer #1: Yes

5. Review Comments to the Author

Reviewer #1: Population cycle and it’s causes has been studied in small mammals in the temperate region. This article is an interesting attempt to understand the population cycle in neotropic region in mammal through various sources of available information. The studied species has interesting population dynamics and ecology which has been nicely covered in the article to understand causes of population cycle (density dependent and density independent factors) which has wide scope of correlating and investigating with the aspects such as human ecology, carbon and nutrient cycle. However, there are comments and suggestions to the authors to which will improve the manuscript. Comments and suggestion which can be incorporated as follows:

1. The title should specifically mention the studied species rather using “a large mammal” for more clarity on the manuscript.

2. It will be better if authors will keep the unit of areas uniform in the entire study (Km2 or Hectare)

3. The site specific information on genetic make-up or inbreeding have not been discussed as a factor of disappearance of the population in isolated patches. Authors should take a note on this and incorporate with supporting information in the manuscript.

4. The cumulative impacts of different aspects such hunting, environmental stressors, disease in population cycle should be analyzed with existing dataset and can be discussed. At this state authors are touching upon the different aspects but a cumulative impact on population fluctuation is missing.

5. Authors should provide more information emphasizing on the factors for the recolonization of the species.

6. PLOS authors have the option to publish the peer review history of their article (what does this mean?). If published, this will include your full peer review and any attached files.

Reviewer #1: No

---

## [Author Response · Author response to Decision Letter 0]

24 Aug 2022

Dear Bilal,

After responding to the reviewers’ and editor’s comments, please accept our resubmission of the manuscript “Disappearances and population cycles of a large mammal in Neotropical forests” (PONE-D-21-25351), now titled “Large-scale population disappearances and cycling in the White-lipped peccary, a tropical forest mammal”. Comments and responses are as follows:

Responses to editor:

Response: The questionnaire we applied has been added as supplemental material, labelled Supplemental S1 Table

Response: None of the authors are reporting funding support for this study. 

Response: All data used in the study are already included in the tables and figures. There are no additional data sets to be referenced.

5. We note that Figure 1 in your submission contain map images which may be copyrighted. All PLOS content is published under the Creative Commons Attribution License (CC BY 4.0), which means that the manuscript, images, and Supporting Information files will be freely available online, and any third party is permitted to access, download, copy, distribute, and use these materials in any way, even commercially, with proper attribution. For these reasons, we cannot publish previously copyrighted maps or satellite images created using proprietary data, such as Google software (Google Maps, Street View, and Earth). For more information, see our copyright guidelines: http://journals.plos.org/plosone/s/licenses-and-copyright.

Response: The map in Figure 1 is not proprietary information nor is it copyrighted.

Response to Reviewer

1. The title should specifically mention the studied species rather using “a large mammal” for more clarity on the manuscript.

Response: We have changed the title to reference the species, as follows: Large-scale population disappearances and cycling in the White-lipped peccary, a tropical forest mammal

2. It will be better if authors will keep the unit of areas uniform in the entire study (Km2 or Hectare)

Response: We changed to Km2 in the tables but use hectares throughout the manuscript text.

3. The site specific information on genetic make-up or inbreeding have not been discussed as a factor of disappearance of the population in isolated patches. Authors should take a note on this and incorporate with supporting information in the manuscript.

Response: The reviewer suggested we discuss the existence, or not, of genetic factors that could affect disappearances. We do not understand this suggestion, as we do not believe there are genetic drivers of population cycling in WLP. Therefore, we have not added text to address the suggestion. We have already noted that population declines in forest fragments may become long term disappearance or local extinctions, due to limited resources and lack of connectivity to source areas 

4. The cumulative impacts of different aspects such hunting, environmental stressors, disease in population cycle should be analyzed with existing dataset and can be discussed. At this state authors are touching upon the different aspects but a cumulative impact on population fluctuation is missing.

Response: There were already comments in appropriate sections in the manuscript on potential cumulative impacts, but we have also added a dedicated section on potential cumulative impacts on page 20. 

5. Authors should provide more information emphasizing on the factors for the recolonization of the species.

Response: This is beyond the scope of this study, as the existence of recolonization, and the form it takes, likely varies from site to site and has rarely been documented. We have added a comment concerning this under the section Reappearances on page 21

Please note that additional comments made by the reviewer in the pdf version of the manuscript have been addressed in the manuscript and our responses can be viewed in the Word Track Changes copy.

---

## [Editor Report · Decision Letter 1]

5 Oct 2022

Large-scale population disappearances and cycling in the White-lipped peccary, a tropical forest mammal

PONE-D-21-25351R1

Dear Jose M. V. Fragoso

We’re pleased to inform you that your manuscript has been judged scientifically suitable for publication and will be formally accepted for publication once it meets all outstanding technical requirements.

Kind regards,

Bilal Habib

Academic Editor

PLOS ONE

Additional Editor Comments (optional):

Thanks for revision. The paper is recommended for publication.
---

## [Editor Report · Acceptance letter]

11 Oct 2022

PONE-D-21-25351R1 

Large-scale population disappearances and cycling in the White-lipped peccary, a tropical forest mammal 

Dear Dr. Fragoso:

I'm pleased to inform you that your manuscript has been deemed suitable for publication in PLOS ONE. Congratulations! Your manuscript is now with our production department. 

Kind regards, 

on behalf of

Dr. Bilal Habib 

Academic Editor

PLOS ONE